# Mesozooplankton Selective Feeding on Phytoplankton in a Semi-Enclosed Bay as Revealed by HPLC Pigment Analysis

**Cui Feng [1], Mengqi Han [1], Chenchen Dong [1], Jingyi Jia [1], Jianwu Chen [2], Chong Kim Wong [3] and Xiangjiang Liu [1],***

[1]    College of Fisheries, Huazhong Agricultural University, Wuhan 430070, China; fc047142@163.com (C.F.); ahanmengqi@163.com (M.H.); dc15623555078@163.com (C.D.); jiajy94@163.com (J.J.)
[2]    Yangtze River Fisheries Research Institute, The Chinese Academy of Fisheries Sciences, Wuhan 430223, China; chjw@yfi.ac.cn
[3]    School of Life Science, The Chinese University of Hong Kong, New Territories, Hong Kong, China; chongkimwong@cuhk.edu.hk
*     Correspondence: liuxiangjiang@mail.hzau.edu.cn; Tel.: +86-158-2730-2886

**Abstract:** Mesozooplankton have been known to be important consumers of phytoplankton, and the community plays an important role in removing the primary production in the marine ecosystem. In the present study, mesozooplankton grazing on phytoplankton were studied in situ at two sampling stations (TM4 and TM8) in Tolo Harbour. HPLC analysis showed that diatoms were the dominant phytoplankton in the two stations throughout the year, and contributed on average to over 40% of total phytoplankton biomass. Dinoflagellates were the second most abundant group of phytoplankton in the two monitoring stations, while the contribution of haptophytes, green algae, cyanobacteria, and cryptophytes was negligible. Feeding experiments, combined with HPLC pigment analysis, were conducted to measure mesozooplankton selective feeding on phytoplankton. The results demonstrated that mesozooplankton displayed a clear feeding selectivity for phytoplankton in Tolo Harbour. Firstly, mesozooplankton showed strong preference for the phytoplankton with the size of 20–200 μm, which suggested that the grazing selectivity and grazing rates of mesozooplankton were affected by the size of the food particles. On the other hand, mesozooplankton assemblages in Tolo Harbour displayed significant feeding selectivity for diatoms, dinoflagellates, and cryptophytes over other types of phytoplankton. The three algae groups are all the major phototrophic components in marine planktonic communities, and they often cause red tides in the marine environment. These results, taken together, suggested that mesozooplankton should play an important role in the regulation of red tides.

**Keywords:** mesozooplankton; grazing rate; feeding selectivity; HPLC; phytoplankton

## 1. Introduction

The mesozooplankton (0.2–2 mm) community plays an important role in the pelagic food web, which includes numerous taxonomic groups such as copepods, cladocerans, tunicates, larvae of marine invertebrates, and noctilucales. Copepods are usually the dominant mesozooplankton in marine ecosystems. They are key components of the typical marine food chain, transferring materials and energy from primary producers to higher trophic levels such as fish, invertebrates, sea birds, and marine mammals [1,2]. They are also the major predators of protozoans, linking the microbial food web to the typical food chain [1]. They could also increase the export fluxes of carbon and nutrients from the light transmittance zone through the passive deposition of their large fecal particles [3,4] and

the active transport of substances to depth during diel vertical migrations [5,6]. These functions in mesozooplankton are largely dependent on their grazing rate and feeding selectivity.

Mesozooplankton are generally size-selective feeders and omnivorous in most areas that the feeding selectivity of mesozooplankton helps to shape the structure of microbial communities. In oceans, microzooplankton and mesozooplankton together can consume an average of 56.9% of oceanic primary production and 34% of coastal primary production. Although in most cases mesozooplankton herbivory is smaller than that of microzooplankton [7], it still contributes a significant loss of primary production [8,9]. On the other hand, mesozooplankton, as predators of microzooplankton, may make an indirect effect on phytoplankton production and composition [4,10]. The omnivorous feeding of mesozooplankton eventually contributes to stabilizing the food web components in responding to food limitation, quality deficit, climate forces, and anthropogenic stresses [11,12].

Traditional methods for studying the feeding selectivity of herbivorous zooplankton usually include (i) applying microscopic cell counting to study the number variation characteristics of phytoplankton cells before and after grazing [13], (ii) gut content examination using light microscopes and scanning electron microscopes (SEM) [14], (iii) use of radioisotope tracers [15], and (iv) use of fluorescent tracers such as chlorophyll *a* (Chl-*a*) [16]. However, each of these methods has a certain limitation. Recently, high-performance liquid chromatography (HPLC) pigment analysis technology has been applied for the study of phytozooplankton [17], and the in situ feeding selectivity of zooplankton [18,19]. The use of a chromatographic method based on HPLC has allowed the rapid separation and accurate quantification of over fifty chlorophyll and carotenoid pigments in a single operation [20]. Phytoplankton pigments have been shown to be valuable chemotaxonomic markers of phytoplankton taxa [21]. For example, peridinin is an unambiguous marker of dinoflagellates [22]. Marker pigments have also been used to identify small and fragile microalgae that are frequently lost in microscopic techniques through destruction by preservatives. A previous study showed the significant relationships between the taxon-specific pigment concentrations and the taxon-specific cell numbers in southwestern Black Sea [23]. Similarly, Wong and Wong et al. also found that a significant correlation was found between peridinin concentration and dinoflagellates density and between the concentration of fucoxanthin and the density of diatoms in Tolo Harbour [24]. Tolo Harbour is a landlocked bay with a large number of algal blooms and red tides in the northeastern part of Hong Kong [25,26]. Although Hong Kong has adopted a series of measures to effectively control water pollution in Tolo Harbour, the phytoplankton biomass and the number of recorded red tides are still relatively large compared with Mirs Bay [27]. Tolo Harbour accounted for about 40% of the red tide incidence recorded in Hong Kong in the last four decades. The common blooming taxa include *Noctiluca scintillans*, *Karenia mikumotoi*, *Akashiwo sanguinnea*, *Takayama tuberculata*, *Scrippsiella trochoidea*, *Thalassiosira tealata*, and *Chaetoceros salsugineum*. So far, the frequency of algal bloom events in inner and middle Tolo Harbour is still around 25–41% after sewage abatement. Meanwhile, previous studies showed that a copepod community in Tolo Harbor was always dominated by small copepods such as *Paracalanus* and *Oithona*, but the impact of mesozooplankton grazing in bloom occurrence has not been clearly conducted.

In the present study, using Tolo harbour as the model, we try to examine the mesozooplankton selective feeding on phytoplankton in a semi-enclosed bay. Firstly, we tried to examine the seasonal and spatial variability of the phytoplankton and mesozooplankton community in Tolo Harbour. Then, we evaluated the grazing rates and feeding selectivity of mesozooplankton on phytoplankton of different taxonomic groups and size fractions.

## 2. Materials and Methods

### 2.1. Field Observations and Sample Treatments

Field surveys and sample collections were conducted at two different sites located in inner Tolo Harbor (TM4, 22°26′ N; 114°13′ E, water depth ~6 m) and outer Tolo Channel (TM8, 22°29′ N; 114°18′ E,

water depth ~20 m) in the northeastern part of Hong Kong (Figure 1). From March 2011 to May 2012, sampling was conducted at approximately bimonthly intervals. The physical parameters including salinity, temperature, dissolved oxygen (DO), and pH at the surface (0.5 m) were measured in situ using a YSI Multiparameter Water Quality Sonde 6600 V2-2 (Xylem Analytics, Hemmant, Australia). Surface seawater was collected at ~0.5 m using a Van Dorn sampler for nutrient and biological analyses, and the water samples were filtered through a 200-μm mesh to remove mesozooplankton. Then, using a Whatman GF/C (1.2 μm) filter (Whatman, Maidstone, UK), the filtrate for the pigmentation analysis filter were stored at −80 °C until pigment extraction, and the filtrates samples for nutrient analyses were stored at −20 °C. Concentrations of the total inorganic nitrogen (TIN), phosphate ($PO_4^{3-}$), and silica ($SiO_2$) were analyzed with a SKALAR Continuous Flow Analyzer (SKALAR Analytical, Breda, Netherlands).

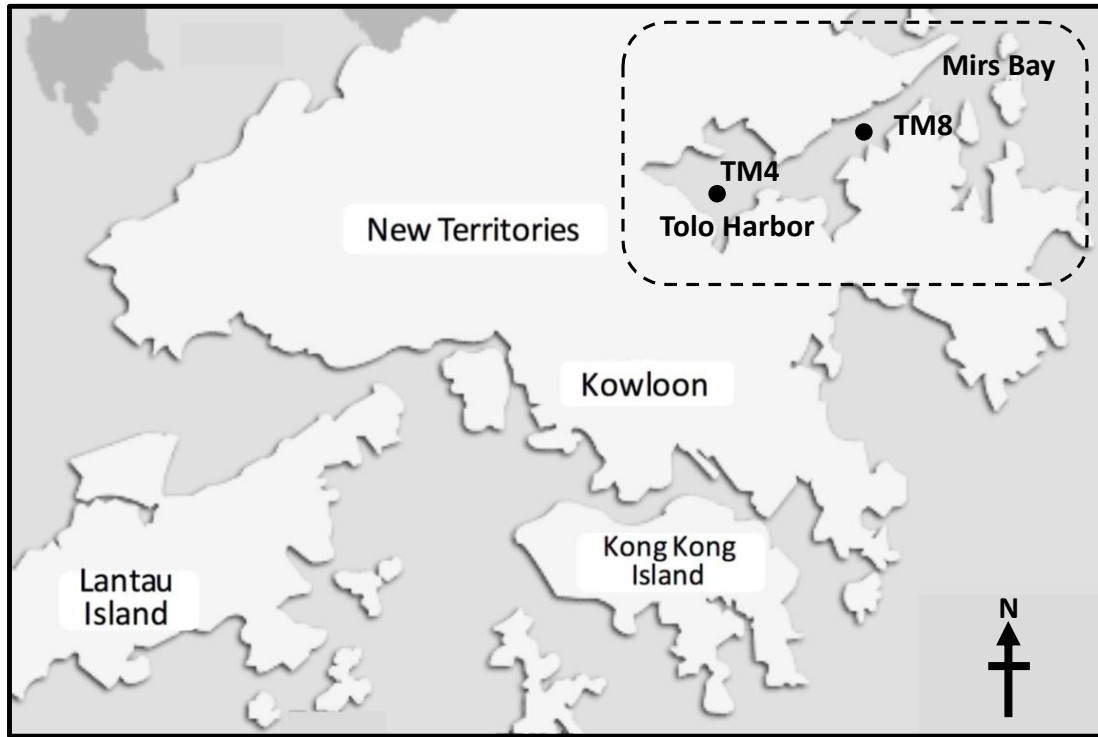

**Figure 1.** Location of sampling sites in inner Tolo Harbour (TM4: 22°26′ N; 114°13′ E) and Tolo Channel (TM8: 22°29′ N; 114°18′ E).

Each seawater sample was preserved with Lugol's iodine (0.5% final concentration) for 200 mL of phytoplankton samples. Phytoplankton and ciliates were concentrated by settling. Diatoms, dinoflagellates, prymnesiophytes, cryptomonads, and chlorophytes were identified, and the counts were performed at 400× using an inverted microscope (Leica Microsystems, Wetzlar, Germany). Ciliates were counted and identified by the Utermohl method at 200× magnification with a Leica DM IL inverted microscope as described by Chen et al. [28]. Mesozooplankton from the seabed 2 m to the surface were collected by a plankton net of 25# (0.5 m diameter, 167 μm mesh size). Samples were transferred into plastic bottles and adjusted to a volume of 500 mL using 200 μm filtered seawater. Mesozooplankton samples were preserved in formalin (4% final concentration) for taxonomic identification and counting under a dissecting microscope.

*2.2. Mesozooplankton Grazing Experiments*

A sufficient number of mesozooplankton living samples were collected from the above-mentioned plankton nets and immediately placed in a cooler filled with seawater as a target for feeding experiments. Ambient seawater for grazing experiments was collected by immersing 20 L polycarbonate carboys

below the water surface. The cooler and carboys were then returned to the laboratory and grazing experiments were conducted within 2 h after the sampling. The surface seawater was filtered through a 200 μm sieve to collect plankton (phytoplankton and small, microzooplankton) below 200 μm as food for mesozooplankton feeding.

Feeding experimental treatment group: the mesozooplankton (50 to 100 mL) were added to 1.2 L polycarbonate (PC) culture bottle, and the same aliquots amount was fixed with formalin (5% final concentration) for the enumeration of mesozooplankton. Control group: no mesozooplankton. Next, both the experimental treatment group and the control group were filled with the surface seawater filtered by the above method, and each group had three parallel samples. By doing this, mesozooplankton abundance was enriched to more than 10 times that in the original seawater. Nutrients (10 μmol/L $NaNO_3$ and 1 μmol/L $KH_2PO_4$) were added to all bottles to promote phytoplankton growth. All the PC bottles were placed in an outdoor incubator and cooled with running seawater for 24 h to simulate a sea feeding process. Size-fractionated Chl-a concentrations were determined at initial (triplicate) and 24 h (duplicate from each bottle) time points.

### 2.3. HPLC Pigment Analysis

In order to conduct pigment analysis, 500 to 1000 mL of seawater was taken from the experimental group and the control group, respectively. All samples were sequentially filtered through 20 and 5 μm poly-carbonate membrane filters (GE, 47 mm diameter) and a glass fiber filter (Whatman, GF/F, 0.7 μm pore size, 47 mm diameter) under low vacuum pressure. The filters were folded in half, blotted dry, and stored at −80 °C until extraction. Before the detection, the phytoplankton-containing filters were cut into small pieces under dim light and transferred to a 15 mL centrifuge tube with 10 mL 90% acetone (HPLC grade, Sigma). The centrifuge tube was wrapped in aluminum foil and treated with ultrasonic vortex for 5 min, and then extracted overnight at −20 °C. After extraction, centrifuge at 4800 rpm for 20 min. The supernatants were filtered through Millipore syringe filters (Hydrophobic, PTFE, 0.2 μM pore size) to eliminate filter and cell debris from the extract. To identify and quantify the phytoplankton pigments, an HPLC (HP Agilent 1100 Series) with an Agilent Eclipse XDB-C18 reversed phase column with a flow rate of 1.0 mL/min was used. Samples were injected into the HPLC system according to method described by Wong and Wong [24]. The HPLC system was pre-calibrated with standard pigments from DHI (Institute of Water and Environment, Hørsholm, Denmark). Comparatively identified photosynthetic pigments included fucoxanthin, peridinin, 19′-hex-fucoxanthin, alloxanthin and Chl-*b*, and Chl-*a*. The HPLC column was run for 30 min to ensure that all pigments were retained. Concentrations of each pigment were calculated based on peak areas in the chromatogram and the equation of the standard curve for each pigment.

### 2.4. CHEMTAX Analysis

The matrix factorization program CHEMTAX was applied to estimate temporal changes in the phytoplankton community structure at the class level at the two stations following the method of Latasa [29,30]. Matrices A to E, artificially generated by Latasa [30], were used to obtain the most feasible initial pigment/Chl-*a* ratios, but But-fuco and haptophytes Type 4 were removed from the calculations in this study because the concentrations of But-fuco, which is present in haptophytes Type 4, were clearly lower than the other pigments. In addition, since prasinoxanthin was not detected but Chl *b* was often found in this study, the term green algae (i.e., prasinophytes and chlorophytes) is used here. The initial pigment/Chl-*a* ratios for green algae in matrices A to E were the same as those of prasinophytes without prasinoxanthin reported by Latasa [30]. Monthly pigment data at the two stations were treated separately for the CHEMTAX analyses. Ten successive runs of CHEMTAX among matrices A to E were made. The convergence of different pigment ratios with the successive runs is directed towards the true values [30]. Therefore, we averaged the final pigment ratios from those that were convergent after the 10 successive runs among the 5 matrices to obtain the most promising initial pigment ratios before incubation at each station. For estimates of phytoplankton community

composition after incubation, the averaged final pigment ratios obtained using the above-mentioned procedures with the monthly pigment data were entered into the seed matrix. We confirmed that the final pigment matrices showed little change even after several further CHEMTAX runs using the monthly pigment data. Therefore, the final pigment matrices mentioned above were simply used as the initial pigment matrices for estimating the selective grazing of mesozooplankton after incubation.

*2.5. Data Analysis*

The phytoplankton growth rate (K, d$^{-1}$) for different size groups or specific phytoplankton taxa in controls ($k_c$) and treatments ($k_t$) were calculated from the equation

$$K = \frac{\ln(P_t/P_0)}{t} \tag{1}$$

where $P_t$ is the final taxon-specific or size-fractioned chl *a* concentration, $P_0$ is the initial taxon-specific or size-fractioned chl *a* concentration, and t is the incubation time in d.

The mesozooplankton clearance (F) and ingestion (I) rates were calculated using the equations of Frost (1972) [31]:

$$F\left(l\ \text{ind}^{-1}\ d^{-1}\right) = \frac{k_c - k_t}{n} \tag{2}$$

$$I\left(\text{ng Chl } a\ \text{ind}^{-1}\ d^{-1}\right) = F \times P' \tag{3}$$

where n (ind·l$^{-1}$) is the number of mesozooplankton that was added to the treatment bottles, and P′ is the mean concentration of chl-*a* during grazing incubation, which can be calculated by the equation P′ = ($P_t − P_0$)/k/t, where t is the time of incubation.

The mesozooplankton feeding selectivity index ($\alpha_i$) was calculated by comparing the frequency distribution of specific prey in the diet and in the environment according to the Chesson selectivity index (CSI) [32,33]:

$$\alpha_i = \frac{r_i}{n_i} \times \frac{1}{\sum_{j=1}^{m} \frac{r_j}{n_j}} \tag{4}$$

where $r_i$ is the ratio of prey category i in the predator's diet, $n_i$ is the ratio of prey category i available in the environment, and m is the number of prey types. Values of index $\alpha_i$ vary between 0 and 1. When $\alpha_i = 1/m$ (i.e., m = 6 $\alpha_i$ = 0.17) indicates no preference for prey i (the prey is consumed according to its availability in the environment), $\alpha_i > 1/m$ (i.e., m = 6 $\alpha_i > 0.17$) indicates that there is preference for prey i, and $\alpha_i < 1/m$ (i.e., m = 6 $\alpha_i < 0.17$) indicates avoidance for prey i.

The three replicates are expressed as the mean ± SD. Significance levels of differences in Chl-*a* concentrations, mesozooplankton ingestion rates, and $\alpha_i$ values were measured using one-way ANOVA analysis (One-factor Analysis of Variance). Further, significant levels of differences in mesozooplankton abundances between two sampling sites were measured using the Mann–Whitney U-test. The differences between groups were considered as significant at *P* < 0.05.

## 3. Results

*3.1. Seasonal Variation of the Surface Physic-Chemical Parameters*

To better elucidate the relationship between the mesozooplankton grazing and physic-chemical characteristics of the waters they inhabit, the water quality parameters and nutrient profiles were recorded at TM4 and TM8 during our sampling. As shown in Figure 2, the variation trend of salinity at TM4 was also similar to that at TM8 from March 2011 to May 2012. The temperature was the highest (31 °C) in August and the lowest in January (16 °C) at both TM4 (Figure 2A) and TM8 (Figure 2B). DO was higher in January (10 mg/L) than that in other months at both TM4 (Figure 2C) and TM8 (Figure 2D). The pH value was lower in January (pH = 7.5) than that in other months at TM4 (Figure 2C), but the pH value at TM8 was always stabilized at 8.0 from March 2011 to

May 2012 (Figure 2D). The concentrations of $PO_4^{3-}$ were always stabilized at a lower level at TM4 (0.52–1.29 μmol/L) (Figure 2E) and TM8 (0.52–2.68 μmol/L) (Figure 2F) during the full year. The TIN in June (25 μmol/L) and March (28 μmol/L) was higher than that in other months at both TM4 (Figure 2E) and TM8 (Figure 2F). The highest concentration of $SiO_2$ was detected in March 2011 (26.62 μmol·$L^{-1}$) at TM4 (Figure 2E), while the $SiO_2$ concentration at TM8 was stable from March 2011 to May 2012 (Figure 2F).

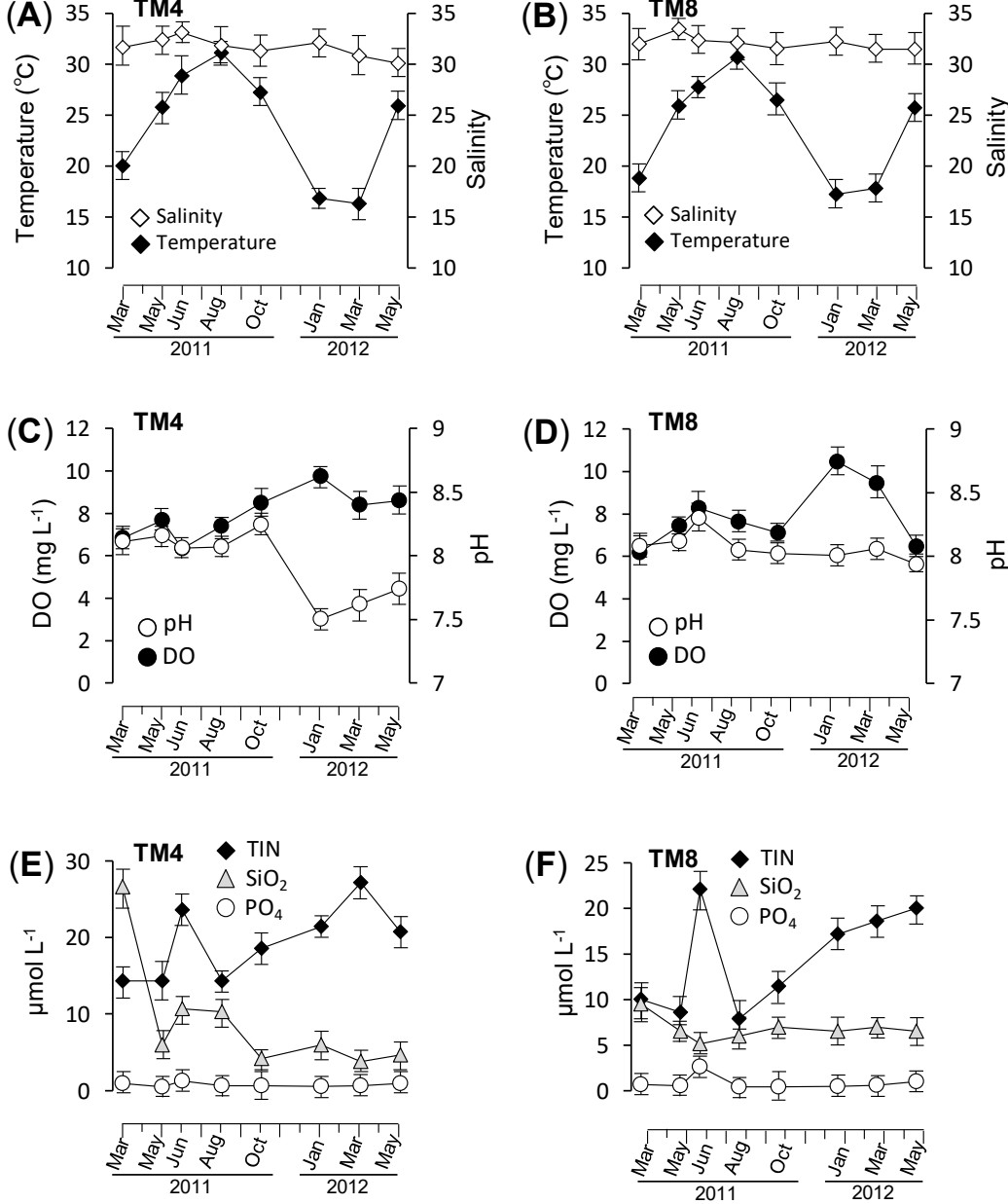

**Figure 2.** Temporal variation in the surface physicochemical parameters measured at the two stations: (**A**,**B**) temporal variations in the surface temperature and salinity; (**C**,**D**) temporal variations in the surface dissolved oxygen (DO) and pH; (**E**,**F**) total inorganic nitrogen (TIN), $PO_4^{3-}$, and $SiO_2$ concentrations.

### 3.2. Seasonal Variation of Phytoplankton Composition and Growth Rates

Phytoplankton densities were consistently higher in TM4 than those in TM8. Averaged over the entire study period, phytoplankton densities were in the order of $10^5$ cells $ml^{-1}$ in TM4 and $10^3$ cells $ml^{-1}$ in TM8. Diatoms represented the most dominant taxa in both TM4 and TM8, the diatoms *Pseudo-nitzchia*,

*Chaetoceros*, *Leptocylindrus*, and *Skeletonema* were common in both sites. Dinoflagellates common at both sites included *Prorocentrum*, *Heterocapsa*, *Karenia*, and *Scrippsiella*. Cryptophytes were also commonly found in both sites, but their densities (average: $297 \pm 144$ cells ml$^{-1}$ in TM4; $56 \pm 85$ cells ml$^{-1}$ in TM8) were extremely low compared with those of diatoms. Phytoplankton community compositions in terms of absolute biomass and relative contributions to total Chl-*a* at the two sampling sites are shown in Figure 3. The Chl-*a* concentrations at TM4 (4.33–16.15 µg/L) were significantly higher than that at TM8 (1.55–4.07 µg/L) from March 2011 to May 2012 ($P < 0.05$) (Figure 3A). At TM4, the highest value of Chl-*a* was observed in August 2011 (16.15 µg/L), when the bloom occurred. The phytoplankton at TM4 were dominated by cells in the intermediate size fraction (5–20 µm) in June, August, and October (bloom period such as *Takayama tuberculata*, *Scrippsiella trochoidea*, *Cyclotella choctawhatcheeana*). In other months, however, phytoplankton at TM4 were dominated by the largest size fraction (20–200 µm). At TM8, the Chl-*a* concentration showed weak variations during the full year (from 1.5 µg/L in June 2011 to 4.0 µg/L in January 2012). Similar to TM4, phytoplankton at TM8 were dominated by the largest size fraction (20–200 µm) except in August.

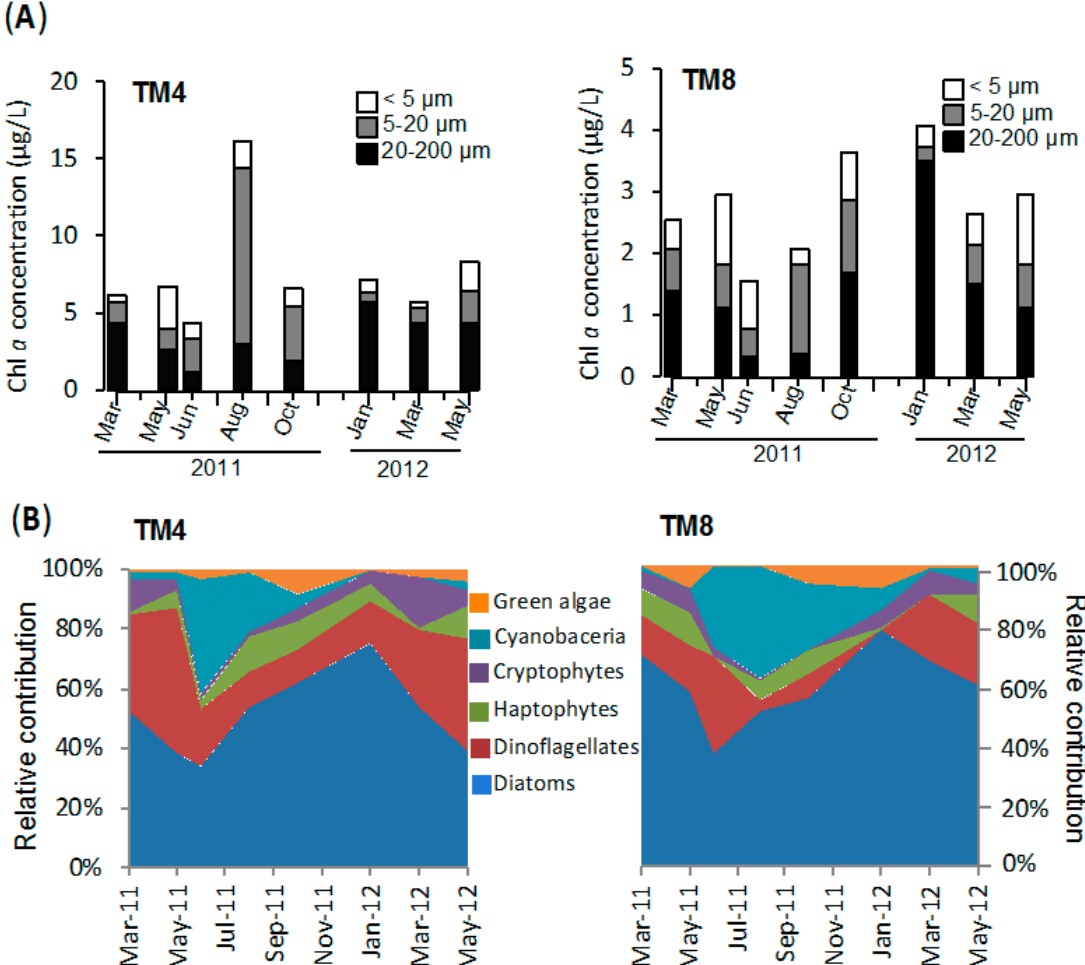

**Figure 3.** Seasonal variation in phytoplankton composition. (**A**) Temporal variations in chlorophyll *a* concentration for phytoplankton in the <200, 20−200, 5−20, and <5 µm size fraction at TM4 and TM8. (**B**) The relative contribution of different phytoplankton groups to total Chl-*a* at TM4 and TM8.

Using HPLC, the relative contribution of different phytoplankton groups to Chl-*a* at TM4 and TM8 was similar in many aspects (Figure 3B). (1) The diatoms group was the most dominant component of the phytoplankton during the full year at both TM4 (35–70%) and TM8 (36–78%); the lowest values were 35% (at TM4) and 36% (at TM8) in May 2011, the highest values were 70% at TM4 and 78% at

TM8 in January 2012, respectively. (2) Dinoflagellate was the second most dominant phytoplankton group at TM4 (10–40%) and TM8 (2–30%). (3) The green algae were the least dominant component of the phytoplankton at both TM4 (1–10%) and TM8 (0–8%). (4) The cyanobacteria biomass was showed high variations during the full year. The highest concentrations were observed in June 2011 at TM4 (40%) and TM8 (38%), the lowest values were observed in January 2012 (0%) at TM4 and March 2012 (0%) at TM8.

As shown in Figure 4, pigment-specific phytoplankton growth rates (g) were always positive. The growth rates of diatoms (Figure 4A), cryptophytes (Figure 4B), and dinoflagellates (Figure 4C) were all detected to be higher than that in haptophytes (Figure 4D), green algae (Figure 4E), and cyanobacteria (Figure 4F). The highest growth rate of diatoms (3.53 d$^{-1}$) was recorded during a diatoms bloom in August at TM4 (Figure 4A). The seasonal variation in phytoplankton growth rates for all pigments was similar among the three size fractions at TM4 and TM8. The highest growth rates were recorded from May to September, and the lowest value was determined in January, which was consistent with the temperature variation.

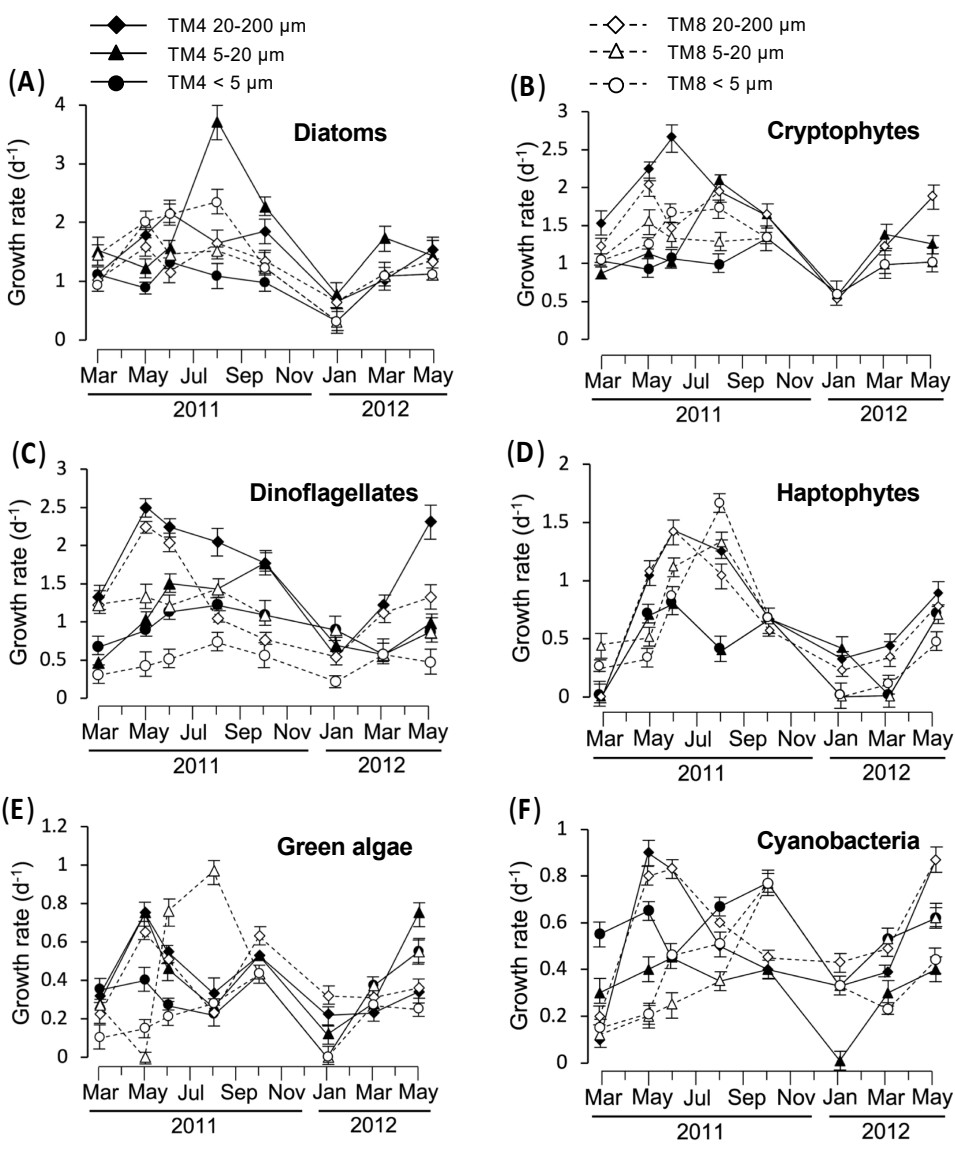

**Figure 4.** Temporal variations in phytoplankton growth rates (g) for Diatoms (**A**), Cryptophytes (**B**), Dinoflagellates (**C**), Haptophytes (**D**), Green algae (**E**) and Cyanobacteria (**F**) in the 20–200, 5–20, and <5 μm size fractions.

### 3.3. Mesozooplankton Community Composition

Abundances of total mesozooplankton, including large heterotrophic dinoflagellate noctilucales (*Noctiluca scintillans*), were also not significantly different between the two stations ($P > 0.05$). Total mesozooplankton ranged from 6312 to 19,055 ind·m$^{-3}$ at TM4 (Supplemental Table S1) and from 5135 to 12,914 ind·m$^{-3}$ at TM8 (Supplemental Table S2). In addition, *Noctiluca scintillans* blooms were observed during early spring (January to March) at TM4 (Supplemental Table S1).

Among the total mesozooplankton, copepods were the most abundant group, followed by cladocerans and gastropod larvae at TM4 (Supplemental Table S1), and barnacle larvae and bivalve larvae at TM8 (Supplemental Table S2). Cladoceran *Penilia. avirostris* and *Pseudevadne tergestina* occurred in the study area throughout the year (Table 1). Average densities of *P. avirostris* at the two stations varied from 350 ind·m$^{-3}$ in March to <5 ind·m$^{-3}$ during the summer from June to October (Table 1). Copepod communities at two stations were dominated by small body-length species (<1 mm) of the genera *Oithona* and *Paracalanus*.

**Table 1.** Comparison of annual average abundance (with the range in parenthesis) of major groups of total metazoans at two stations (TM4 and TM8) in Tolo Harbour.

| | Abundance (ind·m$^{-3}$) | | Percentage (%) | |
|---|---|---|---|---|
| | **TM4** | **TM8** | **TM4** | **TM8** |
| Copepods | 7116 (4423–17,220) | 5729 (3002–11,751) | 79.6 (62.1–90.8) | 84.8 (60.2–95.6) |
| Cladocerans | 594 (0–2710) | 70 (10–140) | 6.6 (0–7.2) | 1.0 (0.1–2.2) |
| Tunicates | 130 (8–422) | 82 (10–322) | 1.4 (0.1–5.0) | 1.2 (0.2–4.6) |
| Barnacle larvae | 210 (0–655) | 256 (0–765) | 2.4 (0–8.1) | 3.7 (0–12.4) |
| Decapod larvae | 128 (0–432) | 72 (0–204) | 1.4 (0–3.3) | 1.1 (0–2.9) |
| Bivalve larvae | 282 (0–866) | 205 (0–566) | 3.2 (0–7.8) | 3.0 (0–12.5) |
| Gastropod larvae | 183 (0–787) | 100 (0–487) | 6.6 (0–10.5) | 1.5 (0–9.2) |
| Polychaete larvae | 18 (0–82) | 14 (0–62) | 0.2 (0–0.4) | 0.2 (0–0.4) |
| Jellyfish larvae | 41 (0–310) | 24 (0–160) | 2.6 (0–8.5) | 0.4 (0–3.0) |
| Chaetognaths | 237 (0–552) | 201 (0–652) | 2.6 (0–6.6) | 3.0 (0–4.7) |

### 3.4. Mesozooplankton Ingestion Rates for Three Phytoplankton Size Fractions

The seasonal variation in mesozooplankton ingestion of different phytoplankton size fractions and the relative contribution of the three phytoplankton size fractions to total ingestion by mesozooplankton are presented in Figure 5. The ingestion rates of mesozooplankton for large phytoplankton (20–200 μm) were significantly higher than that for the 5–20 and <5 μm size fractions at both TM4 (Figure 5A) and TM8 (Figure 5B) ($P < 0.05$). At TM4, the ingestion rates of mesozooplankton for large (20–200 μm) phytoplankton ranged from 21.19 to 69.78 ng Chl-*a* ind$^{-1}$d$^{-1}$, corresponding to 68–75% of the total ingestion (Figure 5A). Furthermore, the highest ingestion rates of mesozooplankton for the 20–200 (69.78 ng Chl-*a* ind$^{-1}$d$^{-1}$) and 5–20 μm size fractions (26.49 ng Chl-*a* ind$^{-1}$d$^{-1}$) were both recorded in June 2011 (Figure 5A). At TM8, large phytoplankton (20–200 μm) comprised about 60–90% of the phytoplankton ingested by mesozooplankton, with the value of 8.29–57.13 ng Chl-*a* ind$^{-1}$d$^{-1}$ (Figure 5B). The highest ingestion of mesozooplankton for 20–200 μm was determined in October 2011 (57.13 ng Chl-*a* ind$^{-1}$d$^{-1}$), and the lowest ingestion value was detected in January 2012 (8.29 ng Chl-*a* ind$^{-1}$d$^{-1}$) at TM8 (Figure 5B). However, the ingestion rates for the 5–20 and <5 μm size fractions showed low and weak variation during the full year at both TM4 (Figure 5A) and TM8 (Figure 5B).

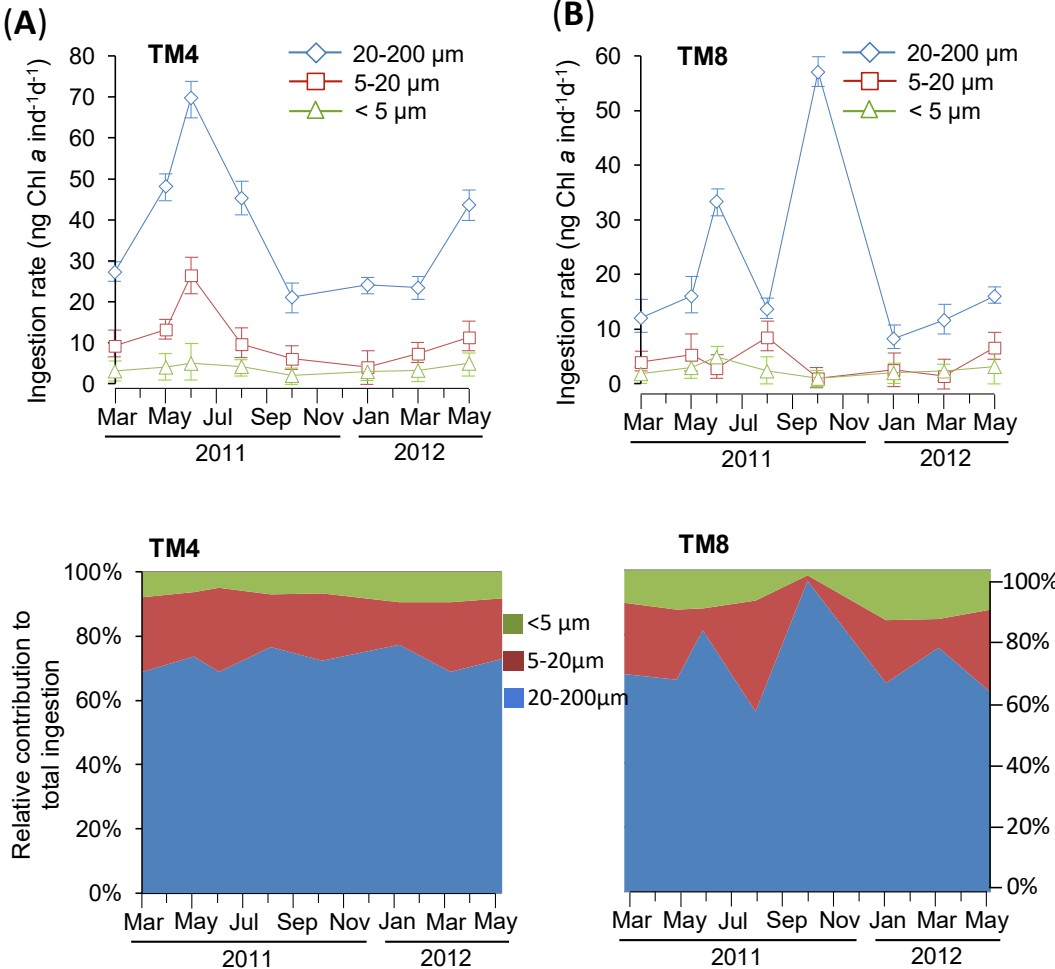

**Figure 5.** Mesozooplankton ingestion rates for three different phytoplankton size fractions. Temporal variations in mesozooplankton ingestion rates for the three phytoplankton size fractions (20–200, 5–20, and <5 μm) and the relative contribution of the three phytoplankton size fractions to total ingestion by mesozooplankton at TM4 (**A**) and TM8 (**B**).

*3.5. Mesozooplankton Ingestion Rates for Specific Phytoplankton Groups*

The seasonal variations in mesozooplankton ingestion of different phytoplankton taxa and the relative contribution of different phytoplankton taxa to the mesozooplankton diet were presented in Figure 6. At TM4, diatoms and cryptophytes account for about 80% of the intake of phytoplankton in mesozooplankton (Figure 6B). The highest ingestion of mesozooplankton for diatoms and cryptophytes was recorded in June 2011, with 30 ng Chl-*a* ind$^{-1}$d$^{-1}$ for diatoms and 31 ng Chl-*a* ind$^{-1}$d$^{-1}$ for cryptophytes (Figure 6A). Dinoflagellates were the other important prey item, accounting for 10–18% of the total ingestion (Figure 6B). The ingestion rates for green algae and cyanobacteria were the lowest for mesozooplankton (Figure 6A).

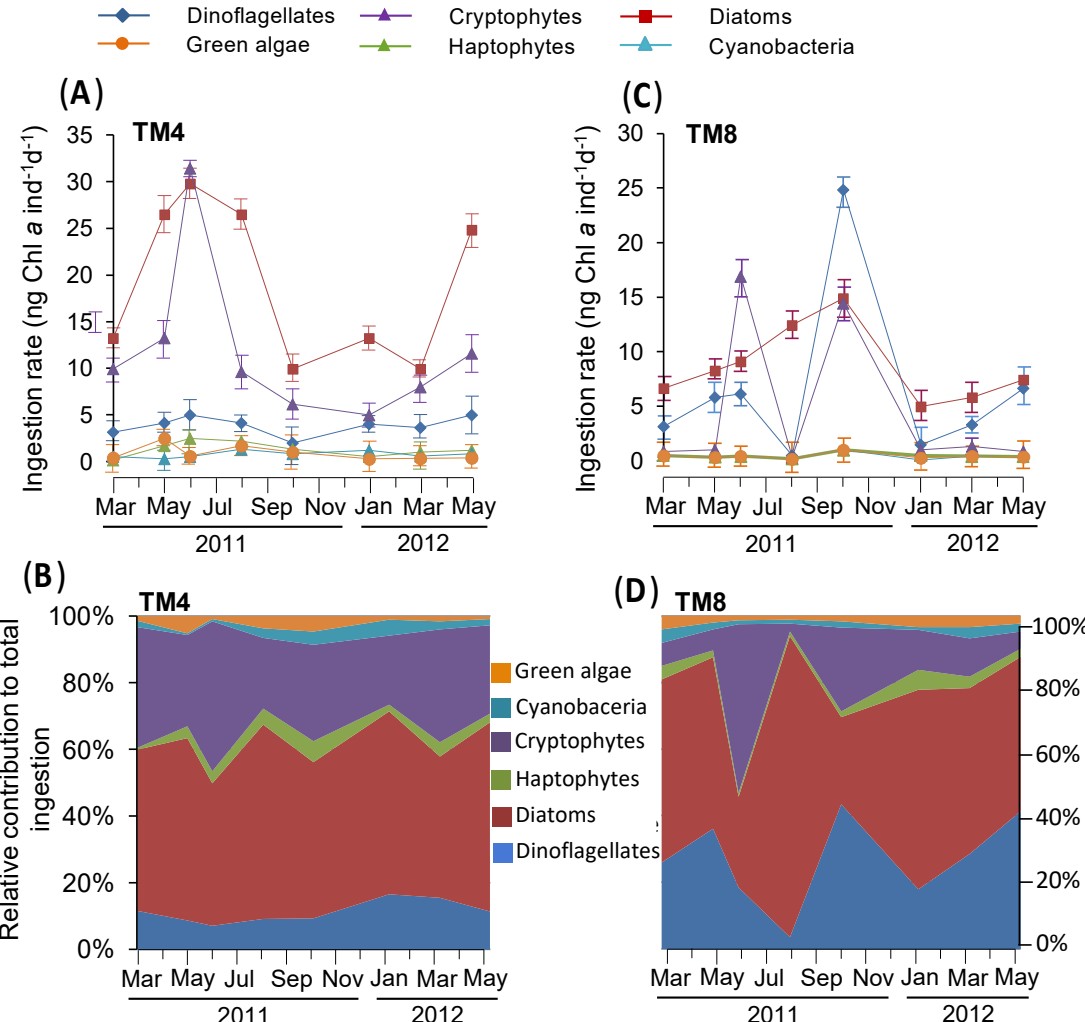

**Figure 6.** Seasonal variations in mesozooplankton ingestion rates for different phytoplankton groups and the relative contribution of different phytoplankton groups to total ingestion by mesozooplankton at TM4 (**A**,**B**) and TM8 (**C**,**D**).

Diatoms and cryptophytes were also the main foods of mesozooplankton at TM8, accounting for about 75% of the total food ingestion (Figure 6D). Meanwhile, in comparison with TM4, dinoflagellates also formed an important food item for mesozooplankton at TM8, except for diatoms accounting for 90% of the total ingestion in August (Figure 6D). Similar to TM4, the total ingestion rates consisted of haptophytes, green algae, and cyanobacteria, which only accounted for about 5% (Figure 6B,D).

*3.6. Mesozooplankton Feeding Selectivity*

The selectivity index αi was used to assess the degrees of mesozooplankton selectivity for specific phytoplankton groups (Figure 7). At TM4, the αi-values for dinoflagellates, cryptophytes, and diatoms were consistently higher than the threshold for selective feeding (Figure 7A), indicating that mesozooplankton had a positive preference for these three phytoplankton groups. The $\alpha_i$-values for haptophytes, green algae, and cyanobacteria were significantly lower than those of the other three phytoplankton groups and were much lower than the selection threshold in most cases ($P < 0.05$). These results indicate that mesozooplankton did not feed effectively on the three kinds of prey.

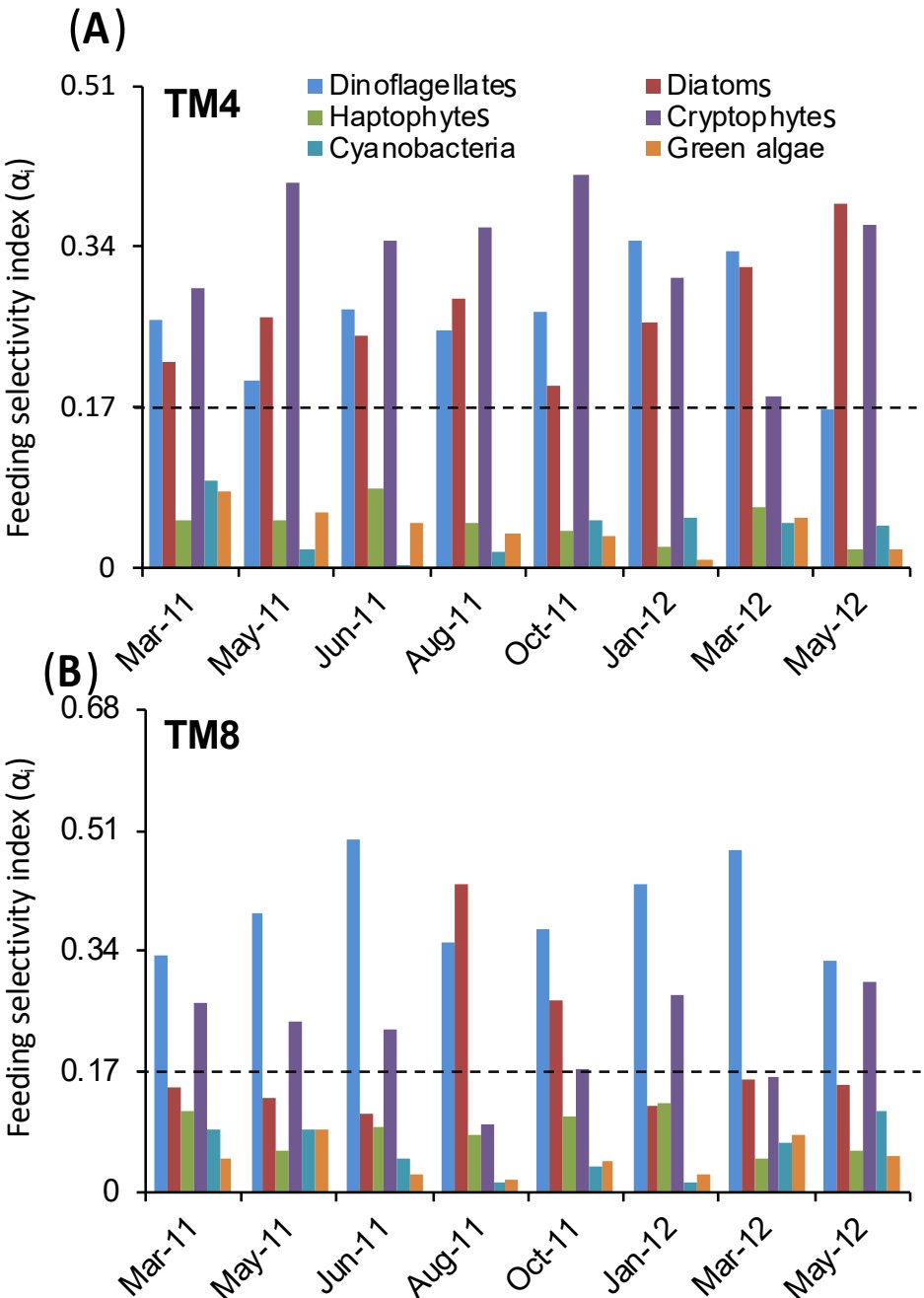

**Figure 7.** Seasonal variations in mesozooplankton feeding selectivity index ($\alpha i$) for different pigments at TM4 (**A**) and TM8 (**B**). The dotted line shows the $\alpha i$-value (0.17) that represents no selectivity. Values above the line indicate preference and values below the line mean avoidance.

Mesozooplankton also exhibited a preference for dinoflagellates and cryptophytes at TM8 (Figure 7B), but the $\alpha_i$-values for cryptophytes were lower at TM8 than that at TM4. Furthermore, contrary to TM4, the $\alpha_i$-values for diatoms were below the threshold except in August and October (diatoms bloom), indicating that mesozooplankton generally had no feeding selection for diatoms at TM8. Similar to TM4, mesozooplankton avoided green algae, haptophytes, and cyanobacteria in most instances at TM8.

## 4. Discussion

In this study, the feeding selectivity of mesozooplankton was studied by HPLC analysis of phytoplankton pigments. Compared with the gut fluorescence method, the HPLC method permits the separation and analysis of a large number of pigments to provide information on the classification of the phytoplankton. The information on the selective feeding of mesozooplankton is important because it provides the necessary insights for selective feeding of mesozooplankton. It has a great influence on the changes of phytoplankton population, and controls the rhythm, scale, and fate of marine primary productivity [34,35] and how mesozooplankton may contribute to shaping the phytoplankton community structure as well as the overall microbial food web [36]. In order to accurately convert the pigment concentration into a real phytoplankton biomass, we directly divided the samples into extracted samples from monthly surveys and samples cultured in grazing experiments, and calculated the ratio of pigment to chl-*a*, respectively. This allowed us to minimize interference from field light and nutrient conditions, species composition, and phytoplankton physiological conditions.

### 4.1. Differences between TM4 and TM8

In the present study, the concentration of Chl-*a* at TM4 was significantly higher than that at TM8, indicating that the concentration of phytoplankton at TM4 was higher than TM8. The analysis of water physic-chemical parameters showed that the numerical changes in hydrography and nutrient profiles were not detected to be significantly different between TM4 and TM8, so the temperature, nutrients, and other physical and chemical factors were not the main reason which lead to a large difference in phytoplankton concentration in the two sampling points. According to the previous description, TM4 is located in the inner Tolo Harbour, in which the water exchange is poor and suitable for the growing and gathering of phytoplankton. On the other hand, TM8 is located at the intersection of Tolo Harbour and Mirs Bay, in which the water exchange is fast [26]. The rapid exchange of water is not conducive to the growing and gathering of phytoplankton. At the same time, due to the fact the measurement of physical and chemical indicators uses surface water, in TM4, which was previously seriously polluted by the coastal factories located in Tolo Harbour, due to long-term accumulation, the bottom of the nutrient of TM4 is relatively higher than TM8.

### 4.2. Mesozooplankton Ingestion of Different Phytoplankton Size Fraction

In the present study, we found that the mesozooplankton showed a strong preference for the items of 20–200 μm. This conclusion could hold even when the uneven representation of these fractions in the original community was taken into account. In addition, we also found that the mesozooplankton at Tolo Harbor preferred to feed diatoms and dinoflagellates, which were all included in the items of 20–200 μm. These results were in agreement with prior literature observations that mesozooplankton ingested food particles from 2.5–100 μm and showed a strong preference for items of 15–75 μm, which included most diatoms [32]. These findings, as a whole, suggested that the grazing selectivity and grazing rates of mesozooplankton were influenced by the size of the food particles, and mesozooplankton may preferentially graze the food that matched the size of their body, rather than those that are too small or too large in size. On the other hand, since microzooplankton are the major consumers of picophytoplankton (<20 μm) [13,28,37], it is likely that the lower clearance rates of these items of <20 μm algae by mesozooplankton could partially be the result of a trophic cascade induced by mesozooplankton predation on microzooplankton. Similar results about mesozooplankton feeding were also reported in other oceanic regions of Hong Kong. Liu et al. (2010) reported that the mesozooplankton population prefers dinoflagellates in excess of other types of phytoplankton, especially in estuarine waters, even though dinoflagellates only contributed to a small proportion of total Chl-*a* in Port Shelter, Hong Kong [9]. Furthermore, in the eutrophic coastal waters where there is sufficient planktonic food, the feeding selectivity of mesozooplankton is affected by factors such as the compositions of predators and the size and quality of the prey [9].

*4.3. Mesozooplankton Ingestion of Specific Phytoplankton Groups*

In this study, diatoms were the most abundant group of phytoplankton and might be the most important food for mesozooplankton all year round, which was consistent with Katechakis's opinion that diatoms were generally considered to be the most abundant food for copepods [38]. However, some studies have shown that diatoms had an adverse effect on the reproduction of copepods. This may be due to the formation of polyunsaturated aldehydes when some diatoms cells were broken [39,40], or the lack of essential nutrients in the body after ingestion of diatoms, such as DHA (docosahexaenoic acid, an essential fatty acid), among others. [41,42]. However, studies have shown that diatoms are considered to be good foods for copepods [43]. Actually, we recorded more avoidance than preference for diatoms by mesozooplankton in our experiments, which seems to be not closely related to any specific diatom species. In addition to the diatoms, we also found that mesozooplankton assemblages in both TM4 and TM8 prefer dinoflagellates. In general, dinoflagellates are considered to be an important food of copepods because of their higher volume-specific organic content than diatoms [44,45]. It has been shown that copepods feeding on dinoflagellate can increase their egg production and survival, as most dinoflagellates contain high amounts of essential fatty acids, such as DHA [11,46]. Cryptophytes are ubiquitous and one of the major phototrophic components in marine planktonic communities [47,48]. They often cause red tides in the waters of many countries [49,50]. Interestingly, in the present study, we found that mesozooplankton showed strong preferences for cryptophytes in Tolo Harbor, with a high ingestion rate and feeding selectivity index. These findings suggested that the mesozooplankton have a high grazing impact on the formation of cryptophytes blooms in Tolo Harbour.

## 5. Conclusions

In the present study, combining the grazing experiments and HPLC pigment analysis, we found that mesozooplankton displayed a clear feeding selectivity for phytoplankton in Tolo Harbour. Firstly, mesozooplankton showed a strong preference for the phytoplankton with the size of 20–200 μm, which suggested that the grazing selectivity and grazing rates of mesozooplankton were influenced by the size of the food particles. On the other hand, mesozooplankton assemblages in Tolo Harbour displayed significant feeding selectivity for diatoms, dinoflagellates, and cryptophytes over other types of phytoplankton. These three algae groups are all the major phototrophic components in marine planktonic communities, and they often cause red tides in the marine environment. These findings indicated that the mesozooplankton have a high grazing impact on the formation of blooms in Tolo Harbor.

**Supplementary Materials:** The following are available online at http://www.mdpi.com/2073-4441/12/7/2031/s1, Table S1: Mesozooplankton composition and abundance (ind.m$^{-3}$) at TM4 of Tolo Harbour, Table S2: Mesozooplankton composition and abundance (ind.m$^{-3}$) at TM8 of Tolo Harbour.

**Author Contributions:** X.L. and C.K.W. conceived the project. C.F., J.J. and J.C. performed the experiments. C.F., M.H. and C.D. contributed to the sample collection. X.L. and C.F. contributed to the manuscript preparation. All authors have read and agreed to the published version of the manuscript.

**Funding:** This research was supported by the Fundamental Research Funds for the Central Universities to X.J.L. (2662019FW007) and National Science Foundation of Hubei Province to X.J.L. (2019CFB493). Financial support was also provided from the Finance Special Fund of Chinese Ministry of Agriculture (Fisheries resources and environment survey in the key water areas of Tibet).

**Acknowledgments:** This article is dedicated to Zemao Gu and Dapeng Li (College of Fisheries, Huazhong Agricultural University) for their genuine interest in training young scientists. We would like to thank Chi Hung Tang (University of Texas at Austin, USA) for his help in the sample collection.

**Conflicts of Interest:** The authors declare no conflicts of interest.

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
