# Peer review of "Mesozooplankton Selective Feeding on Phytoplankton in a Semi-Enclosed Bay as Revealed by HPLC Pigment Analysis"

_water, doi:10.3390/w12072031_

Round 1
Reviewer 1 Report
Authors in this study have presented the use of HPLC to detect feeding selectivity of mesozooplankton on phytoplankton. The study was conducted well and it presents data that supports the interpretations and conclusions.
My overall appreciation is that the study can be published with minor revisions on the following items:
1) All the statistical methods used must be included in the materials and methods section (eg line 183).
2) The authors stated that they identified phytoplankton taxa. Species composition of phytoplankton should be explained in section 3.2. It is important to know, what kind of phytoplankton taxa were responsible for the blooms and red tides, and how the pigments corresponded to species compositions.
3) The authors talked about "metazoan mesozooplankton". Unicellular noctulicales were mentioned as mesozooplankton due to their similiarities in size but they are not metazoan. Therefore, please deleted it from the text.
4) The authors have found high selectivity of mesozooplankton for diatoms. Diatoms, as endosymbiont are often found in the vacuoles in the cell of noctilucan dinoflagellates. Since noctilucales are members of mesozooplankton, the endosimbiont diatomes would increase selectiveness of mesozooplankton for diatoms. Did you check the endosymbionts in the dinoflagellates?
(Haruna Saito, Ken Furuya, Thaithaworn Lirdwitayaprasit, 2006. Photoautotrophic growth of Noctiluca scintillans with the endosymbiont Pedinomonas noctilucae. Plankton and Benthos Research, 1: 97-101)
minor comments:
in line 95: ML probably means mL
in line 98: insert a space after [23].
in line 162: "water quality parameters" should be used instead of "surface hydrography"
in line 163: "detected" should be changed to "recorded"
in line 184: which phytoplankton taxa were blooming
in line 221: change capital "S" in scintillans to small one.
Author Response
Authors in this study have presented the use of HPLC to detect feeding selectivity of mesozooplankton on phytoplankton. The study was conducted well and it presents data that supports the interpretations and conclusions.
Response:
Thank you very much for your help with the paper review.
My overall appreciation is that the study can be published with minor revisions on the following items:
1) All the statistical methods used must be included in the materials and methods section (eg line 183).
Response:
Thanks a lot for your comments. As you suggested, we have added the statistical analysis methods in the material and method section in Line 195-199 listed as below:
“The three replicates are expressed as the mean ± SD. Significance levels of differences in Chl-a concentrations, mesozooplankton ingestion rates and αi values were measured using one-way ANOVA analysis. And significant levels of differences in mesozooplankton abundances between two sampling sites were measured using the Mann-Whitney U-test. The differences between groups were considered as significant at P < 0.05.” (Line 195-199)
2) The authors stated that they identified phytoplankton taxa. Species composition of phytoplankton should be explained in section 3.2. It is important to know, what kind of phytoplankton taxa were responsible for the blooms and red tides, and how the pigments corresponded to species compositions.
Response:
Thank you very much for your suggestions. As you suggested, we have added the information on the species composition of phytoplankton in Line 222-228.
“Phytoplankton densities were consistently higher in TM4 than those in TM8. Averaged over the entire study period, phytoplankton densities were in the order of 105 cells ml−1 in TM4 and 103 cells ml−1 in TM8. Diatoms represented the most-dominant taxa in both TM4 and TM8, the diatoms Pseudo-nitzchia, Chaetoceros, Leptocylindrus, and Skeletonema were common in both sites. Dinoflagellates common at both sites included Prorocentrum, Heterocapsa, Karenia, and Scrippsiella. Cryptophytes were also commonly found in both sites, but their densities (average: 297 ± 144 cells ml−1 in TM4; 56 ± 85 cells ml−1 in TM8) were extremely low compared to those of diatoms.”(Line 222-228)
3) The authors talked about "metazoan mesozooplankton". Unicellular noctulicales were mentioned as mesozooplankton due to their similiarities in size but they are not metazoan. Therefore, please deleted it from the text.
Response:
Thanks for your suggestion. As you suggested, we have deleted “metazoan” in Line 267, 269, 272.
4) The authors have found high selectivity of mesozooplankton for diatoms. Diatoms, as endosymbiont are often found in the vacuoles in the cell of noctilucan dinoflagellates. Since noctilucales are members of mesozooplankton, the endosimbiont diatoms would increase selectiveness of mesozooplankton for diatoms. Did you check the endosymbionts in the dinoflagellates?
(Haruna Saito, Ken Furuya, Thaithaworn Lirdwitayaprasit, 2006. Photoautotrophic growth of Noctiluca scintillans with the endosymbiont Pedinomonas noctilucae. Plankton and Benthos Research, 1: 97-101)
Response:
Thanks for your question. As you mentioned, the dinoflagellate N. scintillans occurs in two forms (Harrison et al., 2011). Red Noctiluca (bloom in red color) is heterotrophic and fills the role of one of the zooplankton grazers in the foodweb. In contrast, green Noctiluca (bloom in green color) contains a photosynthetic symbiont Pedinomonas noctilucae (a prasinophyte), but it also feeds on other plankton when the food supply is abundant. Red Noctiluca occurs widely in the temperate to sub-tropical coastal regions of the world. It occurs over a wide temperature range of about 10℃ to 25℃ and at higher salinities. It is particularly abundant in high productivity areas such as upwelling or eutrophic areas where diatoms dominate since they are its preferred food source. Green Noctiluca is much more restricted to a temperature range of 25℃-30℃ and mainly occurs in tropical waters of Southeast Asia, Bay of Bengal, in the eastern, western and northern Arabian Sea, the Red Sea, and the Gulf of Oman. In subtropical Tolo Harbour, N. scintillans usually become abundant during winter and early spring when temperature was around 20℃ (Liu and Wong, 2006; Zhang et al., 2017), and is responsible for most of the red tides in coastal waters of Hong Kong. Therefore, N. scintillans in our study area is red Noctiluca without endosymbionts.
Harrison, P. J.; Furuya, K.; Glibert, P. M.; Xu, J.; Liu, H. B.; Yin, K.; Lee, J. H.; Anderson, D. M.; Gowen, R.; Al-Azri, A. R., 2011. Geographical distribution of red and green Noctiluca scintillans. Chinese Journal of Oceanology and Limnology, 29(4): 807-831
Liu, X. J.; Wong C. K., 2006. Seasonal and spatial dynamics of Noctiluca scintillans in a semi-enclosed bay in the northeastern part of Hong Kong. Botanica Marina, 49(2): 145-150
Zhang, S. W.; Harrison, P. J.; Song, S. Q.; Chen, M. R.; Kung, H. S.; Lau, W. K.; Guo, C.; Wu, C. J.; Xu, J.; Liu, H. B., 2017. Population dynamics of Noctiluca scintillans during a bloom in a semi-enclosed bay in Hong Kong. Marine Pollution Bulletin, 121(1-2): 238-248.
minor comments:
in line 95: ML probably means mL
Response:
Thank you very much for your help with the paper review. We have corrected “ML” to “mL”. (Line 107)
in line 98: insert a space after [23].
Response:
Thank you very much for your help with the paper review. We have added a space after [23]. (Line 110)
in line 162: "water quality parameters" should be used instead of "surface hydrography"
Response:
Thank you very much for your suggestions. As you suggested, we have corrected “surface hydrography" to "water quality parameters". (Line 203)
in line 163: "detected" should be changed to "recorded"
Response:
Thank you very much for your suggestions. As you suggested, we have corrected “detected" to "recorded". (Line 204)
in line 184: which phytoplankton taxa were blooming.
Response:
Thanks a lot for your comments. Diatom Chaetoceros socialis was considered to be the causative organism for the algal bloom in TM4 in August.
in line 221: change capital "S" in scintillans to small one.
Response:
Thank you very much for your suggestions. As you suggested, we have corrected “S" to "s". (Line 270)
Reviewer 2 Report
This is potentially an interesting and useful study that brings new data on zooplankton feeding in eutrophicated coastal habitats of the tropical zone. However, I have encountered a number of unclear points or methodological and conceptual questions that need to be addressed in the revisions. These points and questions are listed below. In addition, a number of linguistic mistakes were also encountered. These are also included below:
l. 13-14: "and plays an important role" < " and the community plays an important role"
l. 19-20: "while haptophytes, green algae, cyanobacteria, and cryptophytes contribute poorly" < "while contribution of haptophytes, green algae, cyanobacteria, and cryptophytes was negligible"
l. 34: "The mesozooplankton (0.2-2 mm) community play..." < "The mesozooplankton (0.2-2 mm) community plays..."
l. 38: "They are also the major predators of protozoan..." < "They are also the major predators of protozoans..."
l. 34-53: In these two introductory paragraphs, it would be sensible to include some information on the overall taxonomic composition of marine mesozooplankton. Then, you can follow on this information with a more detailed description of mesozooplankton in your studied habitats (i.e. l. 70-72).
l. 61: "of various of algal groups" < "of various algal groups"
l. 59-62: However, what is the effective taxonomic resolution of the HPLC technique? Most importantly, the discrimination of groups with secondary rhodoplasts (diatoms, silicoflagellates, Bolidophyceae/Parmales, haptophytes, dinoflagellates) may not be straightforward. Maybe, you could develop onthis point abit, either here in "Introduction" or later in "Discussion".
l. 66-67: I think that it is crucial to include some information on what has been known on the taxonomic composition of algal blooms and red tides in Tolo Harbour at this point. Without that, the reader is kept in the dark what have you actually been investigating...
l. 74-77: Here, I think it would be more suitable to include a null hypothesis of your study, or, in other words, to state the hypotheses that you were evaluating. Or was it a purely descriptive study? If yes, then it should be clearly stated here, as well.
l. 85-86: "Surface seawater were collected at ~0.5 m of using a Van Dorn sampler for nutrient and biological analyses..." - Fine, but what about the possible effects of vertical heterogeneity of the phytoplankton community? The composition of the phototrophs below the 50 cm upper layer was left completely unstudied. Isn't it possible that some substantial parts of the community might have been "hiding" in the deeper portion of the water column? I think that this must be clearly addressed in the revision.
l. 138: "including" < "included"
l. 137-141: Here, I am missing the detailed information about how the authors transformed the pigment measurement to abundance estimates of individual groups. Obviously, individual pigments greatly differ in their relative concentration in cells. Therefore, careful standardization of the measured pigment concentration values is needed. This must be taken into account and described in detail.
l. 143-146: Here, I really don!t understand the rationale of these analyses. How could the authors discern the "taxon-specific chl a concentration" in seawater samples (where the taxa co-occur in a mixed community)? I would understand that they could measure the decrease in the concentration of the taxon-specific pigments as result of mesozooplankton feeding but "chl a"??
Fig. 3 - "Diatoms" not "Diatom", "Haptophytes" not "Haptophyte", etc.
l. 205: "As shown in Figure 4, pigment-specific phytoplankton growth rates (g) were always positive." - But how is this possible?? This would mean that the amount of phytoplankton constantly increased throughout the studied period! This is, of course, impossible. So, is Fig. 4 produced on the basis of the measurement taken from the controlled zooplankton-free mesocosms? Otherwise, this seems to be an unrealistic outcome of the observations.
l.208: "growth rate of diatom" < "growth rate of diatoms"
l. 257-258: "The highest ingestion of mesozooplankton for diatom and cryptophyte..." < "The highest ingestion of mesozooplankton for diatoms and cryptophytes..."
l. 263: "However, differ from the TM4 situation," - This is unintelligible. Please, rephrase.
l. 282: "was lower" < "were lower"
l. 318-319: "Therefore, the deeper concentration of phytoplankton at TM4 may be relatively high..." - Well, therefore, it might have been a mistake to only focus on the upper-most layer...
l. 323-324: "In the present study, we found that the mesozooplankton showed a strong preference for the items of 20-200 μm." - Does this conclusion hold even when the uneven representation of these fractions in the original community is taken into account? I suppose so but it has to be explicitly stated.
Author Response
1). This is potentially an interesting and useful study that brings new data on zooplankton feeding in eutrophicated coastal habitats of the tropical zone. However, I have encountered a number of unclear points or methodological and conceptual questions that need to be addressed in the revisions. These points and questions are listed below. In addition, a number of linguistic mistakes were also encountered. These are also included below:
Response:
Thank you very much for your help with the paper review.
2). Line 13-14: "and plays an important role" < " and the community plays an important role"
Response:
Thank you very much for your suggestions. As you suggested, we have corrected “and plays an important role" to "and the community plays an important role". (Line 14)
3). Line 19-20: "while haptophytes, green algae, cyanobacteria, and cryptophytes contribute poorly" < "while contribution of haptophytes, green algae, cyanobacteria, and cryptophytes was negligible"
Response:
Thank you very much for your suggestions. As you suggested, we have corrected “while haptophytes, green algae, cyanobacteria, and cryptophytes contribute poorly" to "while contribution of haptophytes, green algae, cyanobacteria, and cryptophytes was negligible ". (Line 19-20)
4). Line 34: "The mesozooplankton (0.2-2 mm) community play..." < "The mesozooplankton (0.2-2 mm) community plays..."
Response:
Thank you very much for your suggestions. As you suggested, we have corrected "The mesozooplankton (0.2-2 mm) community play..." to "The mesozooplankton (0.2-2 mm) community plays...". (Line 34)
5). Line 38: "They are also the major predators of protozoan..." < "They are also the major predators of protozoans..."
Response:
Thank you very much for your suggestions. As you suggested, we have corrected “They are also the major predators of protozoan…" to "They are also the major predators of protozoans...". (Line 39)
6). Line 34-53: In these two introductory paragraphs, it would be sensible to include some information on the overall taxonomic composition of marine mesozooplankton. Then, you can follow on this information with a more detailed description of mesozooplankton in your studied habitats (i.e. l. 70-72).
Response:
Thanks for your suggestion. As you suggested, we listed the information on the overall taxonomic composition of marine mesozooplankton (Line 34-37) and the detailed description of mesozooplankton in Tolo Harbour (Line 81-83).
“The mesozooplankton (0.2-2 mm) community plays an important role in the pelagic food web, which included numerous taxonomic groups such as copepods, cladocerans, tunicates, larvae of marine invertebrates and noctilucales. The copepods were usually the dominated mesozooplankton in the marine ecosystem.” (Line 34-37)
“Meanwhile, previous studies showed that copepod community in Tolo Harbor was always dominated by small copepods such as Paracalanus and Oithona, but the impact of mesozooplankton grazing in bloom occurrence has not been clearly conducted.” (Line 81-83)
7). Line 61: "of various of algal groups" < "of various algal groups"
Response:
Thank you very much for your comments. This sentence has been replaced in the revised manuscript. (Line 66-76)
8). Line 59-62: However, what is the effective taxonomic resolution of the HPLC technique? Most importantly, the discrimination of groups with secondary rhodoplasts (diatoms, silicoflagellates, Bolidophyceae/Parmales, haptophytes, dinoflagellates) may not be straightforward. Maybe, you could develop on this point a bit, either here in "Introduction" or later in "Discussion".
Response:
Thanks a lot for your comments. As you suggested, we have added some information on the the effective taxonomic resolution of the HPLC technique. (Line 62-72)
“The use of chromatographic method based on HPLC have allowed rapid separation and accurate quantification of over fifty chlorophyll and carotenoid pigments in a single operation [20]. Phytoplankton pigments have been shown to be valuable chemotaxonomic markers of phytoplankton taxa [21]. For example, peridinin is an unambiguous marker of dinoflagellates [22]. Marker pigments have also been used to identify small and fragile microalgae that are frequently lost in microscopic techniques through destruction by preservatives. Previous study showed the significant relationships between the taxon-specific pigment concentrations and the taxon-specific cell numbers in southwestern Black Sea [23]. Similarly, Wong and Wong et al. also found that significant correlation was found between peridinin concentration and dinoflagellates density and between the concentration of fucoxanthin and the density of diatoms in Tolo Harbour [24].” (Line 62-72)
9). Line 66-67: I think that it is crucial to include some information on what has been known on the taxonomic composition of algal blooms and red tides in Tolo Harbour at this point. Without that, the reader is kept in the dark what have you actually been investigating...
Response:
Thanks a lot for your comments. As you suggested, we have added the taxonomic composition of algal blooms and red tides in Tolo Harbour in Line 76-79.
“Tolo Harbour accounted for about 40% of the red tide incidence recorded in Hong Kong in the last four decades. The common blooming taxa included Noctiluca scintillans, Karenia mikumotoi, Akashiwo sanguinnea, Takayama tuberculata, Scrippsiella trochoidea, Thalassiosira tealata, and Chaetoceros salsugineum.” (Line 76-79)
10). in line 221: change capital "S" in scintillans to small one.
Response:
Thank you very much for your comment. We have changed capital “S” in scintillans to small one. (Line 78, 267, 269)
11). 74-77: Here, I think it would be more suitable to include a null hypothesis of your study, or, in other words, to state the hypotheses that you were evaluating. Or was it a purely descriptive study? If yes, then it should be clearly stated here, as well.
Response:
Thanks a lot for your suggestion. The present study is a purely descriptive study. As you suggested, we have added the description in Line 84-85.
“In the present study, using the Tolo harbour as the model, we try to examine the mesozooplankton selective feeding on phytoplankton in a semi-enclosed bay.” (Line 84-85)
12). 85-86: "Surface seawater were collected at ~0.5 m of using a Van Dorn sampler for nutrient and biological analyses..." - Fine, but what about the possible effects of vertical heterogeneity of the phytoplankton community? The composition of the phototrophs below the 50 cm upper layer was left completely unstudied. Isn't it possible that some substantial parts of the community might have been "hiding" in the deeper portion of the water column? I think that this must be clearly addressed in the revision.
Response:
Thanks a lot for your comments. Tolo Harbour is a shallow inner bay, with a mean depth of 6–7 m in the northeastern corner of Hong Kong. Although Hong Kong has adopted a series of measures to effectively control water pollution in Tolo Harbour, the phytoplankton biomass and the number of recorded red tides are still relatively large. So far, the frequency of algal bloom events in inner and middle Tolo Harbour is still around 25-41% after sewage abatement. So the water transparency in Tolo Harbour only varied 1.5 to 2.6 m. The most plankton were lived in the surface water, so the surface seawater was collected at ~0.5 m in the present study.
13). 138: "including" < "included"
Response:
Thank you very much for your suggestions. As you suggested, we have corrected “including" to "included". (Line 150)
14). Line 137-141: Here, I am missing the detailed information about how the authors transformed the pigment measurement to abundance estimates of individual groups. Obviously, individual pigments greatly differ in their relative concentration in cells. Therefore, careful standardization of the measured pigment concentration values is needed. This must be taken into account and described in detail.
Response:
Thank you for your comments. As you suggested, we have added the detailed description in the Methods (Line 154-174).
“2.4. CHEMTAX analysis
The matrix factorization program CHEMTAX was applied to estimate temporal changes in phytoplankton community structure at the class level at the two stations following the method of Latasa [29,30]. Matrices A to E, artificially generated by Latasa [30], were used to obtain the most feasible initial pigment: Chl-a ratios, but But-fuco and haptophytes Type 4 were removed from the calculations in this study because the concentrations of But-fuco, which is present in haptophytes Type 4, were clearly lower than the other pigments. In addition, since prasinoxanthin was not detected but Chl b was often found in this study, the term green algae (i.e. prasinophytes and chlorophytes) is used here. The initial pigment: Chl-a ratios for green algae in matrices A to E were the same as those of prasinophytes without prasinoxanthin reported by Latasa [30]. Monthly pigment data at the two stations were treated separately for the CHEMTAX analyses. Ten successive runs of CHEMTAX among matrices A to E were made. The convergence of different pigment ratios with the successive runs is directed towards the true values [30]. Therefore, we averaged the final pigment ratios from those that were convergent after the 10 successive runs among the 5 matrices to obtain the most promising initial pigment ratios before incubation at each station. For estimates of phytoplankton community composition after incubation, the averaged final pigment ratios obtained using the above-mentioned procedures with the monthly pigment data were entered into the seed matrix. We confirmed that the final pigment matrices showed little change even after several further CHEMTAX runs using the monthly pigment data. Therefore, the final pigment matrices mentioned above were simply used as the initial pigment matrices for estimating the selective grazing of mesozooplankton after incubation.” (Line 154-174)
15). 143-146: Here, I really don!t understand the rationale of these analyses. How could the authors discern the "taxon-specific chl a concentration" in seawater samples (where the taxa co-occur in a mixed community)? I would understand that they could measure the decrease in the concentration of the taxon-specific pigments as result of mesozooplankton feeding but "chl a"??
Response:
Sorry for causing your misunderstanding. As you suggested, we have added the detail description in the Methods (Line155-169).
“The matrix factorization program CHEMTAX was applied to estimate temporal changes in phytoplankton community structure at the class level at the two stations following the method of Latasa [29,30]. Matrices A to E, artificially generated by Latasa [30], were used to obtain the most feasible initial pigment: Chl-a ratios, but But-fuco and haptophytes Type 4 were removed from the calculations in this study because the concentrations of But-fuco, which is present in haptophytes Type 4, were clearly lower than the other pigments. In addition, since prasinoxanthin was not detected but Chl b was often found in this study, the term green algae (i.e. prasinophytes and chlorophytes) is used here. The initial pigment: Chl-a ratios for green algae in matrices A to E were the same as those of prasinophytes without prasinoxanthin reported by Latasa [30]. Monthly pigment data at the two stations were treated separately for the CHEMTAX analyses. Ten successive runs of CHEMTAX among matrices A to E were made. The convergence of different pigment ratios with the successive runs is directed towards the true values [30]. Therefore, we averaged the final pigment ratios from those that were convergent after the 10 successive runs among the 5 matrices to obtain the most promising initial pigment ratios before incubation at each station.” (Line155-169)
16). Fig. 3 - "Diatoms" not "Diatom", "Haptophytes" not "Haptophyte", etc.
Response:
Thank you very much for your comments. As you suggested, we have corrected the Figure 3, Figure 6 and Figure 7. (Line 249)
17). Line 205: "As shown in Figure 4, pigment-specific phytoplankton growth rates (g) were always positive." - But how is this possible?? This would mean that the amount of phytoplankton constantly increased throughout the studied period! This is, of course, impossible. So, is Fig. 4 produced on the basis of the measurement taken from the controlled zooplankton-free mesocosms? Otherwise, this seems to be an unrealistic outcome of the observations.
Response:
Thanks for your comments. In Figure 4, the pigment-specific phytoplankton growth rates were calculated using results measured from control group (no mesozooplankton), and phytoplankton biomass in control group was obviously increased after one day incubation without mesozooplankton. That’s why the growth rates were almost positive.
18). Line 208: "growth rate of diatom" < "growth rate of diatoms"
Response:
Thank you very much for your suggestions. As you suggested, we have corrected “growth rate of diatom" to "growth rate of diatoms". (Line 257)
19). Line 257-258: "The highest ingestion of mesozooplankton for diatom and cryptophyte..." < "The highest ingestion of mesozooplankton for diatoms and cryptophytes..."
Response:
Thank you very much for your suggestions. As you suggested, we have corrected “diatom and cryptophyte” to “diatoms and cryptophytes”. (Line 305-306)
20). Line 263: "However, differ from the TM4 situation," - This is unintelligible. Please, rephrase.
Response:
Thanks for your comments. We have rephrased “However, differ from the TM4 situation” into “Meanwhile, in comparison with TM4,”. (Line 311)
21). Line 282: "was lower" < "were lower"
Response:
Thank you very much for your suggestions. As you suggested, we have corrected “was lower” to “were lower”. (Line 330)
22). Line 318-319: "Therefore, the deeper concentration of phytoplankton at TM4 may be relatively high..." - Well, therefore, it might have been a mistake to only focus on the upper-most layer...
Response:
Thank you very much for your comments. Sorry for our shortcoming. We have deleted this sentence in the revised manuscript. (Line 366)
23). Line 323-324: "In the present study, we found that the mesozooplankton showed a strong preference for the items of 20-200 μm." - Does this conclusion hold even when the uneven representation of these fractions in the original community is taken into account? I suppose so but it has to be explicitly stated.
Response:
Thank you very much for your comments. Yes, this conclusion could hold even when the uneven representation of these fractions in the original community was taken into account. As you suggested, we have stated it in Line 369-370.
Reviewer 3 Report
The paper describes the results of research aimed at determining the significance of mesozooplankton as a consumer of phytoplankton in the marine ecosystem. The study was conducted in situ at two sites of different depths and hydrology located in inner Tolo Harbor and outer Tolo Channel in the northeastern part of Hong Kong. These study allowed demonstrate that mesozooplankton shows clear nutritional selectivity, i.e. a strong preference for 20-200 μm phytoplankton and diatoms, dinoflagellates and cryptophytes compared to other types of phytoplankton. I find the results obtained interesting, obtained modern with the help of modern research methods. In addition, these studies are part of a series of analyzes of this type conducted in the coastal waters of the word. For these reasons, I recommend this work for publication. However, I suggest correcting the manuscript in several places.
Remarks:
- “Materials and methods” did not describe how phytoplankton size fractions were obtained in chlorophyll and HPLC analyzes, which were presented, among others in fig. 3a, Fig. 4 (which presents the diversity in the growth rate of taxonomic groups including their size fraction). Have they been verified with the results of counting phytoplankton under an inverted microscope? In the text I did not find a description of the results of phytoplankton studies under an inverted microscope (lines 96-98).
- Line 148 – Frost (1972) is missing from the bibliography.
- The description of the feeding selectivity index (αi), which in literature is also sometimes referred to as the Chesson Selectivity Index (CSI; Chesson 1983), needs to be corrected. In lines 155-158 it is stated that the neutral value αi (which can be described by the symbol αneutral, according to Chesson (1978) means the neutral selection and was described by the formula αneutral = 1/m, where: m is the number of prey types and stages con-sidered) is 0.20. This is due to the assumption that m = 5. Meanwhile, Figure 7 indicates that it is 0.17. The value given in Fig. 7 therefore refers not to 5 but to 6 prey types, i.e. the number of systematic algae groups actually analyzed.
- Mathematical formula No. 2 - The abbreviation of the number of mesozooplankton in the volume unit (ind l-1) should be written as l with -1 in superscript. Similarly in the abbreviation: d-1.
- Line 158 – No dot at the end of the sentence.
- Line 227 – Instead of ‘m-3’, ‘m3’ was written.
- P. 11 – Delete "Figure 7." at the top of this drawing.
- In the whole text, check the absence (e.g. lines 38, 39, 41, 42, 48, 51, 56, ...) and put a space in front of the parentheses in which the literature is quoted.
Author Response
The paper describes the results of research aimed at determining the significance of mesozooplankton as a consumer of phytoplankton in the marine ecosystem. The study was conducted in situ at two sites of different depths and hydrology located in inner Tolo Harbor and outer Tolo Channel in the northeastern part of Hong Kong. These study allowed demonstrate that mesozooplankton shows clear nutritional selectivity, i.e. a strong preference for 20-200 μm phytoplankton and diatoms, dinoflagellates and cryptophytes compared to other types of phytoplankton. I find the results obtained interesting, obtained modern with the help of modern research methods. In addition, these studies are part of a series of analyzes of this type conducted in the coastal waters of the word. For these reasons, I recommend this work for publication. However, I suggest correcting the manuscript in several places.
Response:
Thank you very much for your help with the paper review.
Remarks:
1). “Materials and methods” did not describe how phytoplankton size fractions were obtained in chlorophyll and HPLC analyzes, which were presented, among others in fig. 3a, Fig. 4 (which presents the diversity in the growth rate of taxonomic groups including their size fraction).
Response:
Thank you very much for your comments. As you suggested, we have described how phytoplankton size fractions were obtained in chlorophyll and HPLC analyzes in the revised Materials and methods. (Line 136-139)
“All samples were sequentially filtered through 20 μm and 5 μm poly- carbonate membrane filters (GE, 47 mm diameter) and glass fiber filter (Whatman, GF/F, 0.7 μm pore size, 47 mm diameter) under low vacuum pressure. The filters were folded in half, blotted dry and stored at −80 °C until extraction.” (Line 136-139)
2) Have they been verified with the results of counting phytoplankton under an inverted microscope? In the text I did not find a description of the results of phytoplankton studies under an inverted microscope (lines 96-98).
Response:
Thank you very much for your comments. Yes, we have counted the phytoplankton under an inverted microscope. As you suggested, we have added the results of the phytoplankton in the revised Results (Line 221-227).
“Phytoplankton densities were consistently higher in TM4 than those in TM8. Averaged over the entire study period, phytoplankton densities were in the order of 105 cells ml− 1 in TM4 and 103 cells ml−1 in TM8. Diatoms represented the most-dominant taxa in both TM4 and TM8, the diatoms Pseudo-nitzchia, Chaetoceros, Leptocylindrus, and Skeletonema were common in both sites. Dinoflagellates common at both sites included Prorocentrum, Heterocapsa, Karenia, and Scrippsiella. Cryptophytes were also commonly found in both sites, but their densities (average: 297 ± 144 cells ml−1 in TM4; 56 ± 85 cells ml−1 in TM8) were extremely low compared to those of diatoms.” (Line 221-227)
3) Line 148 – Frost (1972) is missing from the bibliography.
Response:
Thank you very much for your suggestions. We have added this literature in the manuscript and reference. (Line 181, 505)
4). The description of the feeding selectivity index (αi), which in literature is also sometimes referred to as the Chesson Selectivity Index (CSI; Chesson 1983), needs to be corrected. In lines 155-158 it is stated that the neutral value αi (which can be described by the symbol αneutral, according to Chesson (1978) means the neutral selection and was described by the formula αneutral = 1/m, where: m is the number of prey types and stages considered) is 0.20. This is due to the assumption that m = 5. Meanwhile, Figure 7 indicates that it is 0.17. The value given in Fig. 7 therefore refers not to 5 but to 6 prey types, i.e. the number of systematic algae groups actually analyzed.
Response:
Thank you very much for your reminding. We have added “Chesson Selectivity Index (CSI)” into the sentence in line 186-187. And as you mentioned, the number of prey types in Fig. 7 is 6, so we corrected the value of αi (i.e. m = 5 αi = 0.20) into (i.e. m = 6 αi = 0.17). (Line 191-193)
5) Mathematical formula No. 2 - The abbreviation of the number of mesozooplankton in the volume unit (ind l-1) should be written as l with -1 in superscript. Similarly in the abbreviation: d-1.
Response:
Thank you very much for your reminding. We have corrected “l -1” and “d-1” to “l -1” and “d-1”, respectively. (Line 181)
6) Line 158 – No dot at the end of the sentence.
Response:
Thank you very much for your comments. We have added a “.” at the end of the sentence. (Line 193)
7) Line 227 – Instead of ‘m-3’, ‘m3’ was written.
Response:
Thank you very much for your comments. We have corrected “m3” to “m-3”. (Line 224-225)
8) P. 11 – Delete "Figure 7." at the top of this drawing.
Response:
Thank you very much for your suggestions. As you suggested, we have removed "Figure 7." at the top of this drawing.
9). In the whole text, check the absence (e.g. lines 38, 39, 41, 42, 48, 51, 56, ...) and put a space in front of the parentheses in which the literature is quoted.
Response:
Thank you very much for your help with the paper review. We have put spaces in front of parentheses in which the literature is quoted.
Round 2
Reviewer 2 Report
In my view, this is a well prepared revision. At least for my review, the authors responded well to all my points and questions. In addition, they supplemented the text with key new paragraphs in the "Material and Methods" that describe the details of the methodology of their pigment analysis.
In the present state, I see no reason why the revised menuscript couldn't be accepted for the journal.
This manuscript is a resubmission of an earlier submission. The following is a list of the peer review reports and author responses from that submission.